

# Ideas and perspectives: Human impacts alter the marine fossil record

Rafał Nawrot[1], Martin Zuschin[1], Adam Tomašových[2], Michał Kowalewski[3], Daniele Scarponi[4]

[1]Department of Palaeontology, University of Vienna, Vienna, 1090, Austria
[2]Earth Science Institute, Slovak Academy of Sciences, Bratislava, 84005, Slovakia
[3]Florida Museum of Natural History, University of Florida, Gainesville, FL 32611, USA
[4]Alma Mater Research Institute on Global Challenges and Climate Change, Università di Bologna, Bologna, 40126, Italy

*Correspondence to*: Rafał Nawrot (rafal.nawrot@univie.ac.at)

**Abstract.** The youngest fossil record is a crucial source of data documenting the recent history of marine ecosystems and their long-term alteration by humans. However, human activities that reshape communities and habitats also alter sedimentary and biological processes that control the formation of the sedimentary archives recording those impacts. These diverse physical, geochemical, and biological disturbances include changes in sediment fluxes due to the alteration of alluvial and coastal landscapes, seabed disturbance by bottom trawling and ship traffic, ocean acidification and deoxygenation, removal of native species, and introduction of invasive ecosystem engineers. These novel processes modify sedimentation rates, depth and intensity of sediment mixing, pore water saturation state, and preservation potential of skeletal remains – the parameters controlling the completeness and spatiotemporal resolution of the fossil record. We argue that humans have become a major force transforming the nature of the marine fossil record in ways that can both impede and improve our ability to reconstruct past ecological and climate dynamics. A better understanding of the feedback between human impacts on ecosystem processes and their preservation in the marine fossil record offers new research opportunities and novel tools for interpreting geohistorical archives of the ongoing anthropogenic transformation of the coastal ocean.

## 1 Introduction

The Holocene and Anthropocene fossil and sedimentary archives (geohistorical records) are an indispensable source of data on ecosystem and climate states preceding the onset of ecological monitoring and instrumental records, allowing reconstruction of long-term human impacts on marine ecosystems (Dietl and Flessa, 2011; Kidwell, 2015; Yasuhara et al., 2020; Dillon et al., 2022). As with any historical record, the geohistorical data are most valuable when their information quality is well understood. However, humans are altering not only marine ecosystems but also the nature of the stratigraphic archives that record those changes (Wilkinson, 2005; Tyrrell, 2011; Oberle et al., 2016; Syvitski et al., 2022). In other words, humans affect not only what is preserved but also how it is preserved because key ecosystem processes affected by human impacts, such as bioturbation and remineralization of organic matter, also control the burial and preservation of skeletal remains. Although, the resulting taphonomic and stratigraphic signatures may pose a challenge for accurately reconstructing



past ecological and climate dynamics, they can also pinpoint historical shifts in ecosystem functioning (e.g., Gooday et al., 2009; Yasuhara et al., 2019; Tomašových et al., 2021) and thus improve our understanding of the consequences of global change.

Here, we propose a conceptual framework for understanding how human alteration of marine ecosystems changes the
completeness and spatiotemporal resolution of geohistorical records forming in marginal marine, continental shelf and slope environments, where human impacts are concentrated (Halpern et al., 2008). We highlight the implications of this phenomenon for marine paleoecology, conservation paleobiology, and paleoclimatic studies and suggest research strategies that can maximize the information value of the geohistorical data extracted from sediment cores and surface death assemblages (i.e., skeletal remains accumulating in the surface mixed layer of present-day seabeds). By providing an
overview of mechanisms by which humans can modify the incipient fossil record, we hope to encourage a more explicit consideration of these effects during interpretation of marine sedimentary and fossil archives.

## 2 What determines the quality of geohistorical records?

The quality of geohistorical records formed by fossil embedded in sedimentary successions is  determined by their spatiotemporal resolution and completeness. Spatiotemporal resolution corresponds to the extent of spatial mixing and time
averaging of fossil assemblages, i.e., the co-occurrence of remains of organisms that lived at different times and/or places in a single sedimentary layer (depositional resolution sensu Kowalewski and Bambach (2008)). Completeness of the record can be understood both as the completeness of fossil assemblages relative to their source communities (controlled by variability in preservation potential, both within and across taxa), and as the stratigraphic completeness (Sadler, 1981) determined by the duration of hiatuses (stratigraphic resolution sensu Kowalewski and Bambach (2008)). Here, we will primarily focus on
the fossil assemblage completeness, which determines the fidelity of a given eco-environmental variable preserved in the geological record relative to its original signal. For example, a fossil sample limited to thick-shelled specimens varying in age by 3000 years has low completeness and coarse temporal resolution, and thus low fidelity with respect to the original composition of the source living assemblage.

In the marine realm, the completeness and resolution of geohistorical records are primarily controlled by four parameters: 1)
net sediment accumulation rate, 2) depth and intensity of sediment mixing below the seafloor, 3) disintegration rates determined mainly by the pore-water saturation state and bioerosion, and 4) skeletal production and durability. Skeletal production depends on community composition and population dynamics, which control the durability of skeletal remains and the rate at which they enter a death assemblage. Subsequently, sedimentation, sediment mixing, and pore-water chemistry determine whether the remains disintegrate near the sediment-water interface and if and at what rate they undergo
burial to historical layers sealed from mixing, dissolution or erosion (Olszewski, 2004). High sedimentation rates tends to increase both temporal resolution and completeness by reducing exposure time of skeletal remains to mixing and dissolution,





while fast disintegration prevents the accumulation of older remains resulting in low completeness but high temporal resolution.

The net effects of these parameters on the quality of geohistorical archives often depends on complex interactions between them (Fig. 1). For example, changes in sedimentation rate modulate benthic community composition (Thrush et al., 2004) but organisms can also modify sedimentation rates through sediment resuspension, stabilization and production (Meadows et al., 2012). High input of skeletal carbonate, in turn, can buffer pore-water chemistry, reducing carbonate dissolution even in undersaturated conditions (Sulpis et al., 2022). Bioturbation and physical reworking generally reduces the temporal resolution, increases the physical wear of skeletal remains and enhances carbonate dissolution by facilitating sulfide oxidation through pore-water irrigation (Aller, 2014). However, it can also lead to the rapid burial of skeletal remains protecting them from higher skeletal disintegration rates near the sediment surface (Tomašových et al., 2019). Skeletal preservation is also affected by biotic interactions such as bioerosion, durophagous predation, and encrustation (Kidwell and Bosence, 1991).

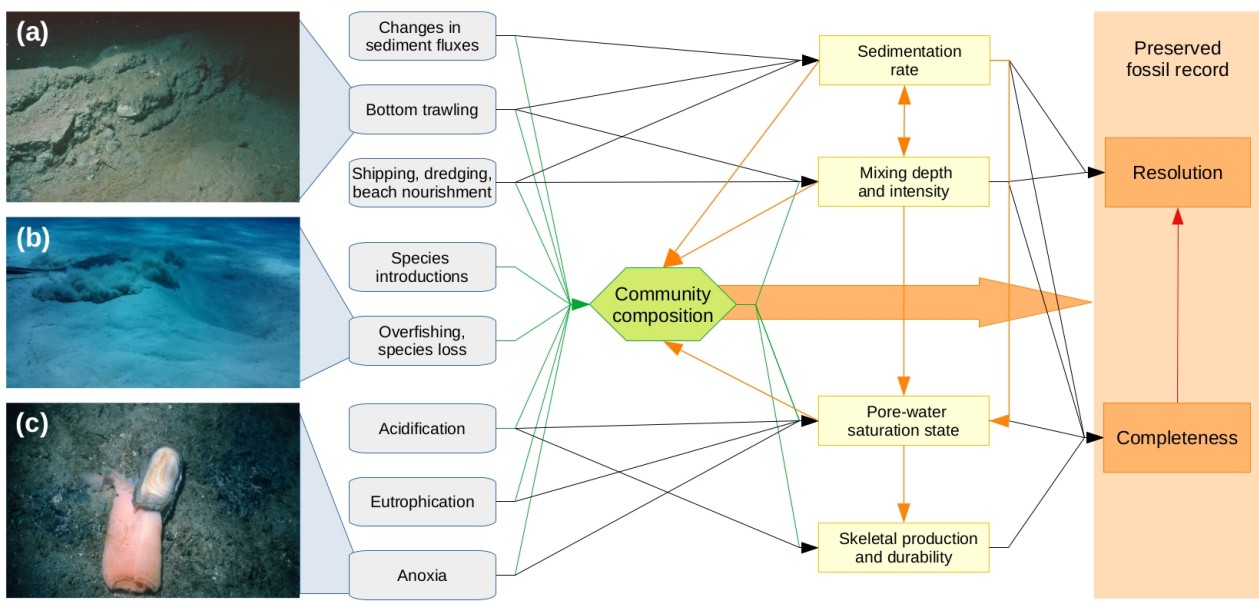

**Figure 1. A conceptual framework for understanding how human impacts can affect the quality of the currently forming marine fossil record with examples of physical (a), biological (b) and geochemical (c) disturbances to marine ecosystems. The four parameters controlling the completeness and resolution of the fossil record (yellow fields) can be affected by human impacts both directly (black arrows) or indirectly through changes in community composition (green arrows) and are linked by complex feedback loops (orange arrows). (a) Disturbance to the seabed (5–10 cm deep furrow) caused by trawling; Gulf of Trieste, northern Adriatic Sea (image courtesy of M. Stachowitsch). (b) Foraging ray causing intensive bioturbation; an older feeding pit, several decimetre in diameter, is visible to the right; San Salvador, Bahamas (image courtesy of M. Kowalewski). (c) A deep infaunal bivalve *Solocurtus* sp., which emerged and died at the sediment surface during a benthic mass mortality event in September 1983 triggered by hypoxia; Gulf of Trieste, northern Adriatic Sea (image courtesy of M. Stachowitsch).**





## 3 Anthropogenic processes altering geohistorical records

The four parameters determining the completeness and resolution of the marine geohistorical records are controlled by natural biological and geological processes driven by tectonics, climate, or biotic interactions. However, anthropogenic processes, which started to operate when humans became a major geological force, can exceed the spatial extent, frequency, rate, and magnitude of the natural drivers (Wilkinson, 2005; Tyrrell, 2011; Syvitski et al., 2022). Human activities can modify the four parameters determining the quality of the record either directly (e.g., elevated sediment supply driven by deforestation can increase sedimentation rates) or indirectly by inducing changes in community composition and ecosystem functioning (e.g., exclusion of burrowers by anoxia results in reduced sediment mixing; Fig. 1).

### 3.1 Physical disturbance

Humans alter the physical environment by changing rates and routes of sediment mobilization, transport, and deposition (Syvitski et al., 2022). On land, these changes are driven by river regulation, cropland and pasture expansion, deforestation, mining, and other landscape alterations. In the sea, the direct physical impacts include exploitation of living resources (trawling), maintenance of global maritime trade (shipping), and interventions mitigating coastal erosion and sea-level rise. These activities affect the magnitude of sediment fluxes to the ocean, coastal geomorphology, sedimentation rates, and seabed properties, leaving clear signatures in stratigraphic sequences (e.g., Martín et al., 2014; Anthony et al., 2014; Handley et al., 2020).

Over the last 3000 years, periods of human population growth and intense landscape transformation were associated with a significant increase in fluvial sediment discharge to marine basins and faster growth of river deltas (Maselli and Trincardi, 2013; Anthony et al., 2014; Syvitski et al., 2022). However, over the last century, despite the continuous increase in sediment production, its delivery to the oceans has declined rapidly due to river damming, reservoir construction, and soil conservation programs (Syvitski et al., 2022). For example, following the construction of the Three Gorges Dam, sediment supply to the Yangtze River Delta declined to the lowest level in its history, while coastal erosion increased (Wang et al., 2018), which should reduce the temporal resolution and completeness of the currently forming geological record (Fig. 2a).

Bottom trawl fishing, seabed excavation, and shipping are drivers of anthroturbation, i.e., anthropogenic sediment mixing (Zalasiewicz et al., 2014; Bunke et al., 2019), in marine ecosystems. Trawling is the most widespread source of physical disturbance to the seabed on continental shelves and slopes (Puig et al., 2012; Amoroso et al., 2018) (Fig. 1a). It can erode, mix and resuspend large volumes of sediment, alter benthic community composition and functioning, and transform seascapes on a scale comparable to agriculture and deforestation on land (Martín et al., 2014; Paradis et al., 2021; Epstein et al., 2022). The impact of trawling on the seafloor ranges from minimal effects to intense winnowing and complete mixing of the top 30 cm (Oberle et al., 2016). For example, in the Baltic Sea, regularly trawled seabeds are thoroughly homogenized down to 25 cm, significantly exceeding the depth of natural hydrodynamic mixing and bioturbation (<10 cm) and erasing the chronological and geochemical signal in sediment cores (Bunke et al., 2019) (Fig. 2b). Trawling-induced fine sediment





resuspension and off-shelf transport can, in turn, lead to a significant increase in sedimentation rates, and thus improve the temporal resolution of the sedimentary record forming on the continental slope below the fishing grounds (Martín et al., 2014).

Ship traffic and anchoring, dredging of navigation channels, and mining of marine sand and gravel can leave an even stronger physical footprint on the seabed, although restricted to shallower waters and more localized than trawling (Schoellhamer, 1996; Rapaglia et al., 2015; Mielck et al., 2021; de Schipper et al., 2021; Watson et al., 2022). For instance, ship wakes resuspend $1.2x10^6$ metric tons of sediments per year in the Venice Lagoon and contribute to the significant erosion of shoals (Rapaglia et al., 2015). Finally, land reclamation, beach nourishment and the construction of coastal

infrastructure to counteract erosion represent a direct large-scale human intervention into sedimentary dynamics both in the coastal zone and in offshore source areas where the sand for these projects is extracted (Mielck et al., 2021; de Schipper et al., 2021).

All these physical disturbances directly impact the parameters controlling the temporal resolution of the geohistorical archives by significantly altering sedimentation rates and increasing the depth of the surface mixed layer. Moreover, mixing-

induced oxygenation of surface sediments increases organic matter remineralization (Epstein et al., 2022) and thus skeletal disintegration rates (Aller, 2014), decreasing the completeness of the currently forming fossil record. Sediment erosion, transport and redeposition triggered by human activities can also decrease spatial resolution through mixing of skeletal assemblages originating from different habitats (Benavente et al., 2005; Bizjack et al., 2017).

### 3.2 Geochemical disturbance

The decline in seawater pH and the saturation state of carbonate minerals triggered by rising atmospheric $CO_2$ directly increases the carbonate dissolution rates (Tyrrell, 2011; Eyre et al., 2018; Fabricius et al., 2020). Consequently, as ocean acidification continues to escalate, the incipient fossil record forming at and directly below the seafloor will dissolve at an increasing pace. Some coral reefs are already undergoing net sediment loss due to dissolution, a pattern expected to accelerate globally (Eyre et al., 2018). More indirectly, acidification affects the ability of calcifying organisms to secrete

their skeletons, which can become thinner, more porous, and fragile (Kroeker et al., 2013). For example, shells of gastropods living near submarine $CO_2$ seeps – a natural system for studying biotic effects of acidification – show high levels of dissolution and reduced mechanical strength (Duquette et al., 2017), which impairs their preservation and long-term accumulation on the seabed (Fig. 2c).

Ocean acidification can interact with anthropogenic eutrophication, reducing carbonate production and preservation rates

(Silbiger et al., 2018). Eutrophication increases primary productivity and reduces light levels, which can induce shifts in benthic community structure and increase the abundance of bioeroders (Lescinsky et al., 2002; Fabricius, 2005; Rice et al., 2020). In nearshore coral reef ecosystems high input of land-derived nutrients reduces coral calcification rates and carbonate sediment production (Chazottes et al., 2008; Silbiger et al., 2018). In contrast, rates of bioerosion can be orders of magnitude higher than in oligotrophic locations, as documented in Hawaiian reefs affected by nutrient-rich submarine groundwater



discharge (Prouty et al., 2017). Moreover, in habitats not subjected to bottom-water oxygen limitation, the eutrophication-driven proliferation of bioturbators can increase oxic-anoxic recycling in the sediment and, thus, carbonate dissolution.

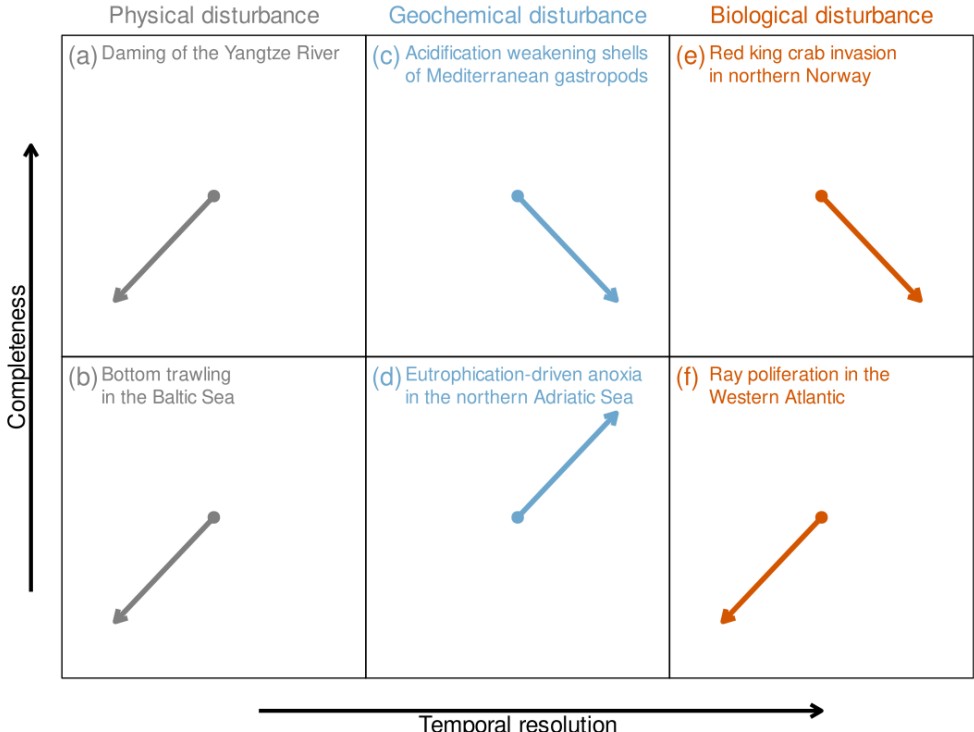

**Figure 2. Examples of expected shifts in the temporal resolution and completeness of the geohistorical record driven by human**
**activities including physical (a-b), geochemical (c-d), and biological disturbances (e-f). For most examples, quantitative estimates of changes in the resolution and completeness of the fossil record following a given disturbance are not available but the orientation of the arrows conveys the expected direction of the shifts in the nature of the fossil record following a given disturbance. (a) Decrease in sedimentation rate on the Yangtze River Delta due to sediment retention by dams (Wang et al., 2018). (b) Mixing of surface sediments in regularly trawled areas of the SW Baltic Sea (Bunke et al., 2019). (c) Degradation of shell**
**mechanical integrity in gastropods exposed to elevated $pCO_2$ near submarine $CO_2$ seeps north of Sicily, which represent a natural experiment on the long-term effects of ocean acidification (Duquette et al., 2017). (d) Decline in bioturbation and sediment ventilation due to seasonal anoxia in the northern Adriatic Sea (Tomašových et al., 2018). (e) Increase in shell-crushing predation and decrease in bioturbation caused by the red king crab invasion in northern Norway (Oug et al., 2018). (f) Increase in bioturbation and shell destruction due to proliferation of cownose ray possibly triggered by a decline in shark abundance in the**
**northwester Atlantic (Myers et al., 2007).**

Eutrophication, however, can also result in hypoxia and anoxia (Rabalais et al., 2014). Under low-oxygen conditions, the emergence of stressed infauna ceases particle reworking long before the actual mortality occurs (Riedel et al., 2014) (Fig. 1c), and the subsequent decline in abundance or extirpation of burrowers limits bioturbation and bioirrigation (Schaffner et
al., 1992; Bianchi et al., 2021) modifying early diagenetic pathways (Middelburg and Levin, 2009). The resulting decline in sediment ventilation by O2 decreases carbonate dissolution by limiting reoxidation of reduced metabolites (Aller, 2014).





Therefore, the increasing frequency of hypoxic events (Rabalais et al., 2014) can improve both the completeness (increased shell preservation) and temporal resolution (reduced sediment mixing) of the Anthropocene record (Fig. 2d; Fig. 3).

### 3.3 Biological disturbance

Humans alter marine ecosystems by introducing non-native species, removing native taxa, or changing their relative abundances. Such widespread impacts often represent secondary consequences of physical and geochemical disturbances. If they affect ecosystem engineers or keystone species (Meadows et al., 2012; Guy-Haim et al., 2018; Bianchi et al., 2021), the resulting changes in bioturbation, sediment geochemistry, or even sedimentation rates (through biologically-mediated sediment resuspension or stabilization) can rapidly alter the nature of the incipient fossil record.

The spread of non-native species can have profound sedimentologic and taphonomic repercussions. For example, the invasion of the Barents Sea by red king crab (*Paralithodes camtschaticus*) feeding on burrowing invertebrates resulted not only in the intensification of shell-crushing predation but also in the decline of sediment reworking and bioirrigation, as indicated by the reduced thickness of the oxidized surface mixed layer (Oug et al., 2018). These functional changes should lead to a decrease in the preservation of skeletal benthos while simultaneously increasing the temporal resolution of their
fossil record (Fig. 2e). In contrast, the proliferation in the Baltic Sea of the non-native polychaetes *Marenzelleria* spp., which burrow much deeper than native species, leads to more effective sediment reworking and bioirrigation (Kauppi et al., 2018), enhancing organic matter remineralization and facilitating the switch from a seasonally hypoxic system back to a normoxic one (Norkko et al., 2012).

Human activities can also result in the regional decline or extirpation of species that directly or indirectly influence
sedimentary and taphonomic processes. For example, trophic cascades driven by overexploitation of large predators can have unexpected consequences for the biotic controls on skeletal preservation. In the northwestern Atlantic, a dramatic decline in shark abundance due to fishing pressure triggered a proliferation of cownose rays *Rhinoptera bonasus* (Myers et al., 2007). As this durophagous species excavates large feeding pits (up to 1 m wide, 20–45 cm deep) to find its infaunal prey, increased ray abundance leads to intensive bioturbation and replacement of seagrass beds by unstable sands (Orth,
1975) thus affecting sediment mixing and shell fragmentation (Fig. 2f).

### 4 Interpreting marine geohistorical records in the Anthropocene

These physical, geochemical, and biological disturbances often interact  (e.g., Norkko et al., 2012; Silbiger et al., 2018; Handley et al., 2020; Geraldi et al., 2020), and their net effect on the quality of the youngest geohistorical archives is context-specific. For example, the negative impact of bottom trawling on the continental shelf records can be accompanied
by improved temporal resolution on the adjacent continental slope due to a resuspension-driven increase in sedimentation rates. The intensity and spatial extent of human alteration of marine sedimentary sequences also vary significantly – from global or regional drivers (e.g, acidification, changes in sediment fluxes) to highly localized ones (e.g., anchoring) –

EGUsphere
Preprint repository

producing a complex mosaic on the seascape, with strongly distorted portions of the continental shelf neighboring areas with a minimal impact (Oberle et al., 2016; Paradis et al., 2021). Moreover, the consequences of global warming and accelerating
sea-level rise, such as increased denudation and coastal erosion (Syvitski et al., 2022; de Schipper et al., 2021), changes in distribution of ecosystem engineers (Aronson et al., 2015; Bianchi et al., 2021; Weinert et al., 2022), or spread of non-native species (McKnight et al., 2021) can intensify many anthropogenic processes that affect the quality of the fossil record.

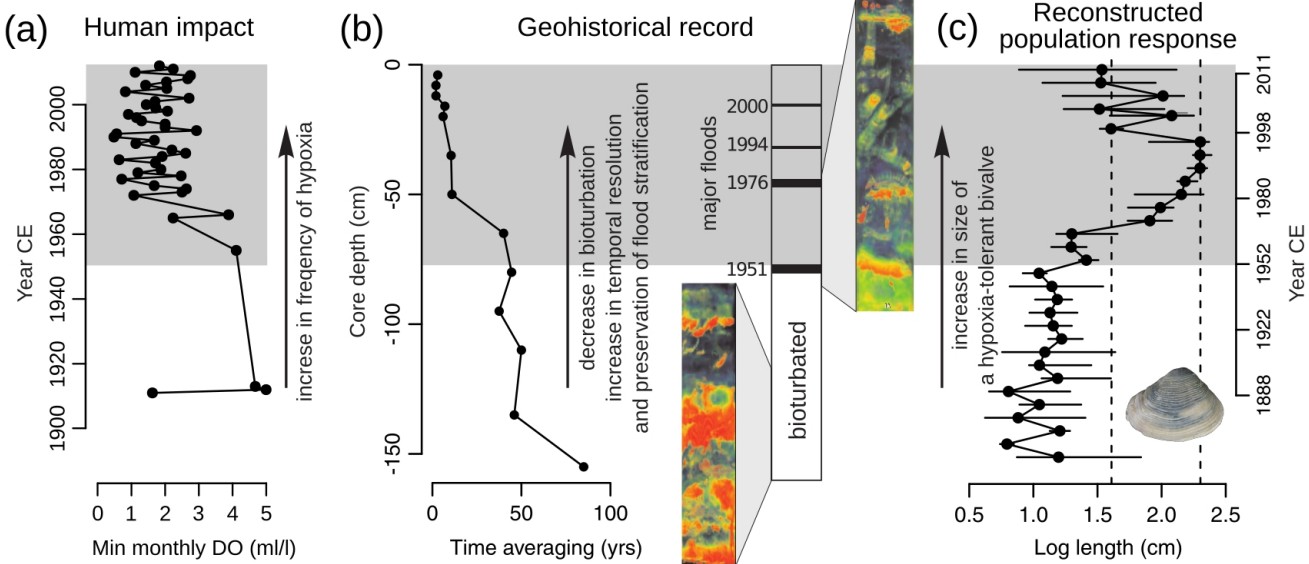

**Figure 3. An example of change in the quality of the Anthropocene fossil record induced by anthropogenic alteration of a marine ecosystem (Tomašových et al., 2018). (a) The 20th-century eutrophication of the northern Adriatic Sea led to decline in dissolved oxygen (DO) concentrations, increase in frequency of seasonal hypoxia and loss of hypoxia-sensitive infauna. (b) In the stratigraphic record of the Po prodelta this regime shift is associated with increased temporal resolution, as expressed by an upcore decrease in bioturbation intensity (strongly homogenized biofabric in the lower part of the core is replaced by discrete**
**traces of echinoids in the upper part as visible on x-ray images), reduced time averaging of molluscan assemblages, and increased preservation of thin silty layers corresponding to decadal floods. Time averaging was measured as interquartile range of post-mortem ages of bivalve shells dated with $^{14}$C-calibrated amino acid racemization methods (30 shells dated per core increment). (b) Fossil assembles from sediment cores document significant increase in abundance and body size of the hypoxia-tolerant bivalve *Varicorbula gibba*, offering a window into pre-impact molluscan community dynamics extending back to the late 19th century –**
**several decades before the the first benthic surveys and over a century before the onset of a systematic ecological monitoring in the northern Adriatic Sea (Tomašových et al., 2018; 2020).**

On the one hand, the interactive effects of humans on both marine ecosystem processes and the nature of the sedimentary archives that record those human-driven shifts (Fig. 1) can represent a challenge for studies relying on geohistorical data.
Changes in the quality of the fossil record coinciding with a major restructuring of ecosystems can mask or exaggerate the magnitude of ecological changes as reconstructed from geohistorical archives. Trawling-induced disturbance can distort the stratigraphic order of top sediment layers, destroy original sedimentary fabric, and truncate, flatten or reverse near-surface geochemical concentration profiles, potentially affecting environmental interpretations based on sediment cores (Oberle et





al., 2016; Martín et al., 2014; Bunke et al., 2019). However, the effects of other factors are more insidious, producing
stratigraphic artifacts that can be mistaken for biological signals. For instance, increased time averaging due to low
sedimentation rates not only reduces the temporal resolution but can also spuriously inflate the evenness and richness of
fossil samples, while decreasing compositional turnover between them (Kidwell and Tomasovych, 2013). Moreover,
sediment mixing and time averaging can obliterate the stratigraphic expression of abrupt regime shifts, making them appear
more gradual (Anderson, 2001; Tomašových et al., 2020), whereas breaks in deposition can turn gradual ecological
transitions into abrupt changes in the stratigraphic record (Kidwell, 1985). Although such effects are most pronounced in the
paleontological data, biogeochemical proxy records, including emerging ones such as biomarkers and sedimentary ancient
DNA, are not immune to the human-induced changes in their preservation (Paradis et al., 2021) and the resolution of
sedimentary archives. Skeletal remains are carriers of many key paleoenvironmental proxies (e.g., stable isotopes of
foraminifera tests), while the interpretation of others ultimately relies on core age-depth models based on radioisotope
profiles or radiocarbon dating of fossils, and can be thus affected by changes in preservation, mixing or sedimentation rates
(Bunke et al., 2019; Kosnik et al., 2015; Lougheed et al., 2018) .

On the other hand, alteration of sedimentary sequences induced by human-driven changes in ecosystem processes offers new
opportunities for geohistorical research addressing different aspects of global change. Anthropogenic signatures in the
completeness and resolution of geohistorical records can serve as essential proxies for major shifts in ecosystem functioning
(Fig. 3). For example, the decrease in bioturbation, increase in organic carbon content, and presence of fine-scale
sedimentary structures like lamination are commonly used as indicators of impacted ecystems and anoxic conditions
(Gooday et al., 2009; Tomašových et al., 2018), while preservation state of skeletal remains observed in sediment cores can
be used to track historical shifts in sediment bioirrigation efficiency (Tomašových et al., 2021) and aid in the identification
of past coral mass mortality events (Wapnick et al., 2004).

Importantly, many human disturbances such as increased sediment supply or coastal hypoxia produce sedimentary records
with higher temporal resolution than those forming under the pre-impact conditions (Fig. 3), facilitating reconstructions of
ecosystem response and recovery. Moreover, even when the quality of the record is reduced, a suite of novel multi-
disciplinary approaches integrating ecological, taphonomic, sedimentological, biogeochemical, and stratigraphic data can
provide robust interpretations of past ecosystem dynamics. Incorporating information on preservation state, time averaging,
and sedimentation rates into paleoecological reconstructions can pinpoint gaps in the record and facilitates disentangling the
effects of major environmental perturbations from spurious patterns arising from changes in preservation or temporal
resolution.

Recent advances in geochronological methods (e.g, Bush et al., 2013; Clark et al., 2014; Gottschalk et al., 2018) allow age
dating of large numbers of individual fossil specimens, providing direct estimates of time averaging, stratigraphic disorder
and hiatus duration (e.g., Kosnik et al., 2007; Scarponi et al., 2013; Nawrot et al., 2022; Tomašových et al., 2022; reviewed
by Tomašových et al., 2023) and thus more realistic assessment of the uncertainties associated with age-depth models
(Kosnik et al., 2015; Lougheed et al., 2018; Dolman et al., 2021). Even more importantly, this approach can be used to





"unmix" fossil assemblages and thus resolve temporal changes in production, body size and community composition, which may not be evident in time-averaged records (e.g., Harnik et al., 2017; Tomašových and Kidwell, 2017; Tomašových et al., 2019, 2020; Clark et al., 2023). Radiocarbon dating of individual fossils can also be combined with analyses of geochemical proxies from the same specimens, allowing extracting well-resolved paleoenvironmental time series from highly time-averaged sedimentary successions (Lougheed et al., 2018).

Finally, numerical simulations that jointly model ecological, sedimentological, and taphonomic processes provide valuable insights into how the taphonomic and stratigraphic overprints can modify biotic and environmental signals. For instance, metacommunity models coupled with preservational processes were used to estimate the magnitude of change in community turnover expected under increasing loss in temporal resolution (Kidwell and Tomasovych, 2013), and the effect of time averaging on the detection of regime shifts in the stratigraphic record was explored by combining numerical simulation with evolutionary models (Tomašových et al., 2020). Contrasting the outputs of such models with the empirical data constrains the range of possible ecological scenarios consistent with the patterns observed in the stratigraphic record.

## 5 Conclusions

Somewhat analogous to the Heisenberg Uncertainty Principle, the very same human-driven processes that have been altering ecosystems also affect our ability to reconstruct these changes based on geohistorical data. The role of humans in transforming the currently forming paleobiological record is already substantial on land (Bennett et al., 2018; Plotnick and Koy, 2020) and will only intensify in the ocean as exploitation of its resources and other human stressors intensify (O'Hara and Halpern, 2022), representing a potential challenge for conservation paleobiology, historical ecology, and paleoclimatic research. At the same time, taking into account stratigraphic signatures left by past human-induced shifts in ecosystem functioning should further augment our ability to use fossils, geochemical proxies, aDNA, and other geohistorical data, to track the history of anthropogenic transformation of the coastal ocean. Therefore, we need a better understating of the effects of human actions on different parameters of the taphonomic equation that control the fidelity of the fossil record in the Anthropocene.

## Author Contribution

R.N. conceived the original idea; R.N., M.Z, A.T, M.K., and D.S. performed literature survey and contributed data; R.N., M.Z, A.T, M.K., and D.S. wrote the manuscript.

## Competing interests

The authors declare that they have no conflict of interest.



## Data availability

No primary data were used in this paper. Data used to create Fig. 3 are available as supplementary materials to Tomašových et al. (2018, 2020).

## Acknowledgments

We thank Michael Stachowitsch for sharing underwater photographs and Tadej Gračner for providing access to the Marine Biology Station Piran, Slovenia, where the initial outline of this paper was formulated.

## Financial support

This work was supported by the NSF (EAR-1559196 and RNC grant EAR-1922562), Slovak Research and Development Agency (APVV-22-0523), Slovak Scientific Grant Agency (2/0106/23), and Ministero dell'Università e della Ricerca 300   (PRIN2022WEZR44).

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
