# Peer review of "Ideas and perspectives: Human impacts alter the marine fossil record"

_EGUsphere, 2023_

## Author Response (AR1)

*In the responses below, when listing relevant changes we updated the line numbers to those in the revised manuscript and marked them in yellow.*

**RC1 RESPONSE:**

We greatly appreciate for the detail review, which helped us to improve the earlier version of our paper. Below we provide a point-by-point summary of the changes we introduced to the manuscript submitted to *Biogeoscineces* and its revised version:

***RC1 – 1. It may be good to explain some terms for wider audience? Such as youngest fossil records, surface death assemblages.***

We specify in the introduction that we focus on the Holocene/Anthropocene fossil record (see also our response to Reviewer 2) and define terms such as surface death assemblages (*line 39*), time averaging (*line 45*) and completeness of the record (*lines 46-49*). Following suggestion of Reviewer 2 in the revised manuscript explicitly refer to the Anthropocene sensu Gibbard et al. (2022) in *line 22*.

***RC1 – 2. p. 2 "The resulting taphonomic and stratigraphic signatures can pinpoint historical shifts in ecosystem functioning (e.g., refs. 10–12)":***
*Yasuhara et al 2019 may be a good review to cite here.*
*Yasuhara et al 2019. Palaeo-records of histories of deoxygenation and its ecosystem impact. In: Laffoley, D., Baxter, J. M. (eds), Ocean Deoxygenation: Everyone's Problem – Causes, Impacts, Consequences and Solutions: 213–224*

We cited Yasuhara et al., 2019 (*line 32*)

***RC1 – 3. p. 2 "…forming on continental shelves and slopes,":***
*Marginal marine environments, too? Such as bays, estuaries, lagoon, etc.*
*You may see Yasuhara et al 2012 review. Most of documented human impacts from paleo records are from such marginal marine environments:*
*Yasuhara et al, 2012. Human-induced marine ecological degradation: micropaleontological perspectives. Ecology and Evolution, 2: 3242–3268.*

We added marginal marine environments to the list (*line 35*).

***RC1 – 4. p. 2 "remains of organisms that lived at different times":***
*As well as places?*

Thank you for pointing this out. We changed the text to "remains of organisms that lived at different times and/or places" (*line 45*).

***RC1 – 5. p. 2 "determined by the duration of hiatuses….ref 14":***
*Sedimentation rate, too, especially in sediment cor studies?*

*In addition to ref 14, Yasuhara et al 2017 is a good review more focused on sediment core based studies.*
*Yasuhara et al 2017. Combining marine macroecology and palaeoecology in understanding biodiversity: microfossils as a model. Biological Reviews: 92, 199–215.*

We refer here (==line 49==) specifically to "stratigraphic resolution" sensu Kowalewski and Bambach (2008). These authors separated the concept of stratigraphic resolution (among strata; related to the duration of diastems separating depositional events that produced the strata) from depositional resolution (within-stratum; controlled by the extent of spatial mixing and time averaging). Of course sedimentation rate affects both aspects of the fossil record.

*RC1 – 6. At the end of p2. I think it's better to be specific what is "the former".*

We changed "the former" to "the fossil assemblage completeness" (==line 50==).

*RC1 – 7. resolution vs completeness:*
*High sedimentation rate leads to high resolution, but how about completeness?*
*Quick postmortem accumulation may lead to better completeness?*
*But dilution can make paleo study difficult in some case?*
*Not sure if this is directly relevant to this story.*

In general, high sedimentation rates tends to increase both temporal resolution and completeness of fossil assemblages, as rapid burial reduces the residence time of skeletal remains in the taphonomic active zone where both mixing and disintegration rates are high. However, dilution of fossils by non-skeletal sediment may indeed complicate some paleoecological studies, especially those focusing on community composition and diversity. Finding sufficient number of individuals when fossil density in very low may require increasing the volume or area of individual samples and thus lead to lower resolution of the resulting data (e.g., when adjacent core increments are combined to increase sample size). Although potentially important, such indirect effects of methodological decisions can vary widely depending on the nature of samples, targeted taxon, depositional setting and goal of any particular study. Thus, we do not address them in our manuscript, as our main focuses is the human alteration of the primary parameters controlling the completeness and resolution of the marine records.

*RC1 – 8. (1)–(4) should have the same names in the first listing and the subsequent paragraphs?*

This part of the manuscript was shorted and rewritten using consistent terms for the four parameters (==lines 56-74==).

*RC1 – 9. "3) Pore-water saturation state" is talking about calcium carbonate fossils and not others, such as slica ones and others like pollen, shark denticles.*
*Also anoxic sediments have another issue, benthos cannot survive and so there is no in-situ fossil records. Virtually all planktons from upper water column?*

*Also, I am not super sure, but I tend to feel anoxic or hypoxic sediment have rather bad fossil preservation (regarding calcium carbonate shells). Because they are organic rich and H2S is changed to H2SO4 with oxygen, leading to calcium carbonate dissolution. But it's more after core recovery, subsampling, and during storage when sediment is exposed to air?*

The paragraph "Pore-water saturation state" was removed from the current version of the manuscript. Although most of our examples are based on carbonate remain, the effects of human actives modifying resolution and completeness of the geological record apply to non-carbonate skeletal elements as well.

Lack of remains of benthic organisms reflecting their earlier disappearance due to persistent anoxic conditions represents true ecological signal. Thus, the fidelity of the fossil record itself is high in this respect but, as in the case of low fossil density driven by high sedimentation rates, it may complicate some paleoecological investigations, for example relying on benthic foraminifera. This, however, depends on specific goals of a study.

Concerning the role of $H_2S$ oxidation in skeletal preservation, this mechanism applies to a situation when organic-rich anoxic sediment is overlain by oxygenated bottom waters and not to persistently hypoxic conditions above the sediment-water interface (as is the case during hypoxic events). Exposure of anoxic/hypoxic sediment to oxygen during sampling may affect to some degree skeletal preservation but this is largely a post-collection problem that can be avoided when samples are processed with caution. We decided that adding a note on this issue is not critical given the focus of this perspective.

*RC1 – 10. Also (1) and (4) are rather related, since sedimentation (rate) is not only by siliciclastic but also biological production (like nanno fossil, planktic foram, and diatom ooze).*

In the new version, we shortly mention interactions between sedimentation rates, community composition and skeletal production (lines 65-68).

*RC1 – 11. Syvitski et al 2009 Sinking deltas due to human activities. Nature Geoscience is also a good ref for the paragraph starting from "Over the last 3000 years," in p. 4.*

There are several excellent papers on this topic by Syvitski et al., we refer here (line 97) to the comprehensive recent review (Syvitski et al. 2022 Nat. Rev. Earth Environ. https://doi.org/10.1038/s43017-021-00253-w).

*RC1 – 12. For physical disturbance, if you consider deep sea. Deep-sea mining and also Ocean Based Climatic Intervention is possible issues.*
*Some refs here:*
*Levin et al 2020 Challenges to the sustainability of deep-seabed mining. Nature Sustainability 3, 10 784-794.*
*Levin et al 2023 Deep-sea impacts of climate interventions: Ocean manipulation to mitigate climate change may harm deep-sea ecosystems. Science: 379, 978–981.*

As noted in the introduction, we restricted our overview to marginal marine, continental shelf and slope environments and thus did not mention physical impacts on the deep-sea record. Compared to these shallower settings, direct human disturbance to strata forming on the abyssal plain is still limited. However, given that Reviewer 2 also suggested adding more information on the deep marine systems, we mentioned the effects deep-sea mining and ocean based climatic intervention in the revised version of the manuscript.

We added (*lines 129-130*): "Finally, emerging industry of deep-seabed mining will extend the impacts of anthroturbation to abyssal plains (Levin et al., 2020)" and include "potential impacts of ocean-based climate interventions (Levin et al., 2023)" when discussing consequences of global warming (*lines 227*).

***RC1 – 13. Can also mention land reclamation, coastal protection, and fish and prawn farming.***

Land reclamation and coastal protection are discussed in the later part of the section on physical disturbance (*lines 126-128*).

***RC1 – 14. Also can mention physical, chemical and biological disturbances are often related each other.***
***For example, Coastal protection may change the distribution of species, resulting biological invasions (Dong et al 2016)***
***Dong et al 2016. The marine 'great wall'of China: local-and broad-scale ecological impacts of coastal infrastructure on intertidal macrobenthic communities. Diversity and Distributions, 22(7), 731-744.***
***Degraded environments by chemical disturbance (eg hypoxia) may have more chance of bio-invasion by tolerant or opportunistic species.***

We discuss interactions between different types of disturbances in the section on the interpretation of the geohistorical records (*lines 217-221*).

***RC1 – 15. P. 6 "changes in the completeness and resolution of the youngest fossil records will also affect the quality of Holocene paleoclimatic and paleoceanographic reconstructions based on them."***
***Do you think such is substantial? If so any example? Substantial such could be mainly limited to the latest Holocene/Anthropocene, that is not really main target of paleoclimatology/paleoceanography? Not critical. Just curious***

This sentence was removed from the current version of the paper. We do think that in some specific cases, effects of human activities may affect the interpretation of paleoclimatic records on continental shelves (although, of course, this concerns only the most recent records). For example, Oberle et al. (2016, p. 129) noted that there is "a large body of work on late Holocene climate reconstructions that is based on upper sediment core data from areas known to be subjected to bottom trawling". Trawling-induced distortion to the stratigraphic order and geochemical

concentration profiles in the uppermost sediment layers can reduce resolution of paleoclimatic proxy records and affect depth-age models used for their interpretation.

Oberle, F. K. J. et al. 2016 Deciphering the lithological consequences of bottom trawling to sedimentary habitats on the shelf, J. Mar. Syst., 159, 120–131, https://doi.org/10.1016/j.jmarsys.2015.12.008

***RC1 – 16. Harnik et al 2017 Assessing the effects of anthropogenic eutrophication on marine bivalve life history in the northern Gulf of Mexico. Palaios***
***is relevant here for the 1st paragraph of p7.***

Thank you for the suggestion. We added Harnik et al. (2017) as one of the examples (*line 270*)

***RC1 – 17. Finally for the title "Human impacts alter their own fossil record".***
***I might misunderstand, but "their own" is not needed? "fossil and geological record" may be better?***

Thank you for the suggestion. The title was modified; in the current version of the manuscript, it reads: "Human impacts alter the marine fossil record".

**RC2 RESPONSE**

Thank you for your positive comments and very helpful suggestions. We discuss them in more detail below.

*RC2 – Line 14: another theme that might be added and briefly explored here is the introduction of the novel materials associated with the Anthropocene. One is concrete, an artificial 'rock' that since the mid-20th century has become abundant around coastlines, making the greatest difference around 'soft-rock' coastlines made of unconsolidated deposits. Another is plastic which, in forms ranging from microplastics to macroplastics of a wide range of shapes and dimensions are already affecting sedimentation both physically, e.g. the work of Russell et al. 2023 Communications Earth & Environment 4 (255) showing how plastics affect dune formation and the review of 'plastistones' by Wang & Hou 2023, Earth Science Reviews 247, 104620, and biologically as now widely explored; shallow-buried plastics will affect redox conditions too.*

That is an excellent point. Although we briefly mention the role of coastal infrastructure, we missed the potential importance of plastic debris.
We inserted the following statement (==lines 131-134==): "In addition to modifying sediment fluxes, humans are also introducing novel materials to sedimentary systems, such as concrete, plastic and other types of marine litter. The presence of micro- and macroplastic debris alters the physical properties of sediments (Russell et al., 2023; Wang and Hou 2023) and interact with marine biota (e.g., Galloway et al., 2017; Hope et al., 2021) affecting sediment stability, erosion, transport and bioturbation rates."

We also modified Fig. 1 by replacing "Changes in sediment fluxes" with "Changes in sediment composition and fluxes", and added the following references:
Galloway, T. S. et al. 2017 Interactions of microplastic debris throughout the marine ecosystem, Nat. Ecol. Evol., 1, 1–8, https://doi.org/10.1038/s41559-017-0116
Hope, J. A. et al. 2021 Microplastics interact with benthic biostabilization processes, Environ. Res. Lett., 16, 124058, https://doi.org/10.1088/1748-9326/ac3bfd
Russell, C. E. et al. 2023 Plastic pollution in riverbeds fundamentally affects natural sand transport processes, Commun. Earth Environ., 4, 1–10, https://doi.org/10.1038/s43247-023-00820-7
Wang, L. and Hou, D. 2023 Plastistone: An emerging type of sedimentary rock, Earth-Sci. Rev., 247, 104620, https://doi.org/10.1016/j.earscirev.2023.104620

*RC2 – Line 22: the Anthropocene perhaps need a reference consistent with the authors' understanding of this new and still informal unit. From the text of this ms, the Anthropocene here seems consistent with the concept of Crutzen as developed stratigraphically, so one of the following might be used:*
*Syvitski, J., Waters, C.N., Day, J. et al. Extraordinary human energy consumption and resultant geological impacts beginning around 1950 CE initiated the proposed Anthropocene Epoch. Commun Earth Environ 1, 32 (2020). https://doi.org/10.1038/s43247-020-00029-y*
*And/or*
*Waters, C.N., Zalasiewicz, J., Summerhayes, C. et al. 2016, The Anthropocene is functionally and stratigraphically distinct from the Holocene. Science 351, DOI: 10.1126/science.aad2622*

This is indeed very contentious topic and we are grateful for highlighting the need for more explicit explanation of what we mean by the Anthropocene. Our understanding of this concept is actually closer to that of Gibbard et al. (2022) and Walker et al. (2024), i.e. as a diachronous and ongoing "event" (or "episode" sensu Head et al. 2022) rather than as a formally defined and globally isochronous chronostratigraphic unit as postulated by e.g., Waters et al. (2016) and Head et al. (2022). Although our conclusions are valid irrespective of which of these two opposing views is accepted, we feel that following diachronous definition of the Anthropocene is consistent with strong spatial variation in the timing and intensity of human modification of the marine fossil record, which is one of the points that we wanted to make in our paper. We also think that this approach better reflects the complex and multifaceted nature of the historical and long-lasting interactions between human beings and the environment.

We admit that our usage of the term was not consistent (both in this and our previous papers), which may lead to confusion. To avoid ambiguity, in the revised manuscript we started the introduction (*lines 26-27*) with "The fossil and sedimentary archives (geohistorical records) of the Anthropocene (sensu Gibbar et al. (2022); see Head et al. (2022) for an alternative view)…" and refer to "post-1950 CE" when describing the time interval (and its stratigraphic expression) corresponding to the Anthropocene understood as an epoch (series) following the proposal of a candidate GSSP by McCarthy et al. (2023) (*line 204*)

Gibbard, P. et al. 2022 The Anthropocene as an Event, not an Epoch, J. Quat. Sci., 37, 395–399, https://doi.org/10.1002/jqs.3416
Head, M. J. et al. 2022 The proposed Anthropocene Epoch/Series is underpinned by an extensive array of mid-20th century stratigraphic event signals, J. Quat. Sci., 37, 1181–1187, https://doi.org/10.1002/jqs.3467
McCarthy, F. M. et al. 2023 The varved succession of Crawford Lake, Milton, Ontario, Canada as a candidate Global boundary Stratotype Section and Point for the Anthropocene series, Anthr. Rev., 10, 146–176, https://doi.org/10.1177/20530196221149281
Walker, M. J. C. et al. 2024 The Anthropocene is best understood as an ongoing, intensifying, diachronous event, Boreas, 53, 1–3, https://doi.org/10.1111/bor.12636
Waters, C. N. et al. 2016 The Anthropocene is functionally and stratigraphically distinct from the Holocene, Science, 351, aad2622, https://doi.org/10.1126/science.aad2622

*RC2 – Line 43: 'fossil' should be 'fossils'*
*Line 96: this might be adjusted to '… mining, urbanization, and other landscape alterations'.*
*Line 123: 106 should be $10^6$ (with superscript)*

Thanks for spotting these issues. All these changes were incorporated into the manuscript

*RC2 – Line 143: the para is good, but very shallow marine-focussed: could add something on effects on oceanic planktonic systems/deep marine systems (thinning of pteropod & foram shells, change in CCD).*

In the revised manuscript we provided additional examples from the deep marine systems (*lines 150-151*): "Deep-sea fossil archives are also affected due to impaired calcification of planktic foraminifera and pteropodes (Bednaršeket al. 2012; Béjard et al, 2023), and shallowing of the calcite compensation depth (Sulpis et al., 2018)".

We also mentioned potential effects of climate interventions on deep marine systems, as suggested by Moriaki Yasuhara (*line 227*).

The following paper were added to the reference list:
Béjard, T. M. et al. 2023 Calcification response of planktic foraminifera to environmental change in the western Mediterranean Sea during the industrial era, Biogeosciences, 20, 1505–1528, https://doi.org/10.5194/bg-20-1505-2023
Bednaršek, N. et al. 2012 Extensive dissolution of live pteropods in the Southern Ocean, Nat. Geosci., 5, 881–885, https://doi.org/10.1038/ngeo1635
Sulpis, O. et al. 2018 Current CaCO3 dissolution at the seafloor caused by anthropogenic CO2, Proc. Natl. Acad. Sci., 115, 11700–11705, https://doi.org/10.1073/pnas.1804250115

***RC2 – Line 265: Perhaps note, too, the effect of the 'bomb spike' from nuclear weapons testing on radiocarbon and other radionuclides, as a factor.***

Nuclear testing added $^{14}$C to the atmosphere, causing a rise in delta $^{14}$C. However, the concentration of bomb $^{14}$C in the upper layers of the ocean changes much slower than in the atmosphere over decades. The time and peak delta $^{14}$C values of oceanic dissolved inorganic carbon vary significantly depending on their location. Thus, bomb $^{14}$C in oceans is of limited significance to high-resolution chronology, unlike atmospheric bomb $^{14}$C (Dutta, 2016).

We modified *lines 271-272* as follows: "Radiocarbon dating of marine fossils and dead remains (notwithstanding the alteration generated by nuclear testing; Dutta, 2016) …".

Dutta, K. 2016 Sun, Ocean, Nuclear Bombs, and Fossil Fuels: Radiocarbon Variations and Implications for High-Resolution Dating, Annu. Rev. Earth Planet. Sci., 44, 239–275, https://doi.org/10.1146/annurev-earth-060115-012333

**DC3 RESPONSE**

Many thanks for your positive feedback and constructive suggestions.

*RC3 – 1. In lines 54 to 64, the four key parameters controlling the completeness and resolution are introduced. This section is provides the theoretical backbone of all further observations, as all anthropogenic impacts modify these parameters, which in turn modify the completeness and resolution. This information in itself is a highly useful synthesis for readers not familiar with the topic. I think it could use some more references to support the very general claim made here, and give the interested reader the opportunity to read up on more details. Adding a figure that visualizes the effect of the four parameters on the examined properties (completeness, resolution) would help sell this much more.*

In the revised manuscript we cited Kidwell (1986) and Kowalewski (1997) to support the statements made in this paragraph (*lines 61-64*), as well as refer to Olszewski (2004), Kowalewski and Bambach (2008), and Tomašových et al. (2019) (already cited elsewhere in the manuscript) when introducing the key parameters (*line 57*).

Added to the reference list:
Kidwell, S. M. 1986 Models for fossil concentrations: paleobiologic implications, Paleobiology, 12, 6–24, https://doi.org/10.1017/S0094837300002943
Kowalewski, M. 1997 The reciprocal taphonomic model, Lethaia, 30, 86–88, https://doi.org/10.1111/j.1502-3931.1997.tb00447.x

The interactions between the key parameters and their effects on completeness and resolution are already shown (in much simplified way) on Fig. 1. Unfortunately, given the short format of "Ideas and perspective" articles, we cannot provide more extensive discussion of the these aspects, so providing additional references where such information can be found is a great idea.

*RC3 -- **Figure 1 should be clarified. Connections between anthropogenic impacts and their control on the 4 key parameters are shown, but it is visually unclear what exactly their effect is. Do they increase/decrease the parameters, stabilize them, etc? What is the causal relationship between them (and are there potential feedback loops)? The same holds for the relationship between the 4 parameters and completeness and resolution. Using clear visual cues (line width, color coding, symbols such as ++ to indicate positive feedback etc) would make the figure much easier to comprehend.***

The interactions between human activities, community composition, sedimentary dynamics and formation of the fossil record are inherently complex. Thus, the main goal of this figure was to provide a very concise, schematic overview of these interactions in order to highlight the multiple pathways in which humans can alter quality of the geohistorical records. However, the direction (i.e. increase/decrease of the affected parameters) and strength of these effects are highly context dependent. For example, bottom trawling can decrease local net sediment accumulation rates by inducing erosion and resuspension of finer particles, but the resuspended material can be transported offshore leading to increase in sedimentation in other parts of the basin. Similarly,

although sediment mixing generally reduces completeness by facilitating skeletal disintegration, deep bioturbation reaching below the surface completely-mixed layer can also lead to rapid burial of shells enhancing their preservation. It is difficult to faithfully represents such complexities on a single figure (or indeed create a generalized model specifying which exact effects are positive or negative). We decided instead to discuss these problems in the main text together with specific examples shown on Fig. 2. We feel that adding symbols like +/- to the few interactions that are rather straightforward (e.g. effect of acidification on pore water saturation), while marking most of the others as "variable" would only decrease the clarity of the figure.

In the revised figure we used more consistent color coding to denote different types of effects and modify the caption as follows (*lines 76-81*): "A simplified conceptual framework for understanding how human impacts can affect the quality of the currently forming marine fossil record with examples of physical (a), biological (b) and geochemical (c) disturbances to marine ecosystems. The four parameters controlling the quality of the fossil record (yellow fields) can be affected by human impacts both directly (blue arrows) or indirectly through changes in community composition (green arrows). The four parameters are linked in turn by complex feedback loops (orange arrows) and together determine the resolution and completeness of the record (black arrows). The direction and strength of the effects of human activities is often context dependent (see examples in the main text)."